# A Portable Servoregulation Controller to Automate CO_2_ Removal in Artificial Lungs

**DOI:** 10.3390/bioengineering9100593

**Published:** 2022-10-21

**Authors:** Navid Shaikh, Andrew Zhang, Jesse Jenter, Brandon Nikpreljevic, John Toomasian, William Lynch, Alvaro Rojas-Peña, Robert H. Bartlett, Joseph A. Potkay

**Affiliations:** 1Extracorporeal Life Support Laboratory, Department of Surgery, University of Michigan, Ann Arbor, MI 48109, USA; 2VA Ann Arbor Healthcare System, Ann Arbor, MI 48105, USA

**Keywords:** respiratory distress, extracorporeal CO_2_ removal, ECMO, closed-loop systems, medical control systems, portable miniaturized systems, servosystems

## Abstract

Artificial lung (AL) systems provide respiratory support to patients with severe lung disease, but none can adapt to the changing respiratory needs of the patients. Precisely, none can automatically adjust carbon dioxide (${\mathrm{CO}}_{2}$) removal from the blood in response to changes in patient activity or disease status. Because of this, all current systems limit patient comfort, activity level, and rehabilitation. A portable servoregulation controller that automatically modulates ${\mathrm{CO}}_{2}$ removal in ALs to meet the real-time metabolic demands of the patient is described. The controller is based on a proportional-integral-derivative (PID) based closed-loop feedback control system that modulates sweep gas (air) flow through the AL to maintain a target exhaust gas ${\mathrm{CO}}_{2}$ partial pressure (target ${\mathrm{EGCO}}_{2}$ or ${\mathrm{tEGCO}}_{2}$). The presented work advances previous research by (1) using gas-side sensing that avoids complications and clotting associated with blood-based sensors, (2) incorporating all components into a portable, battery-powered package, and (3) integrating smart moisture removal from the AL to enable long term operation. The performance of the controller was tested in vitro for ∼12 h with anti-coagulated bovine blood and 5 days with distilled water. In tests with blood, the sweep gas flow was automatically adjusted by the controller rapidly (<2 min) meeting the specified tEG${\mathrm{CO}}_{2}$ level when confronted with changes in inlet blood partial pressure of ${\mathrm{CO}}_{2}$ (p${\mathrm{CO}}_{2}$) levels at various AL blood flows. Overall, the ${\mathrm{CO}}_{2}$ removal from the AL showed a strong correlation with blood flow rate and blood p${\mathrm{CO}}_{2}$ levels. The controller successfully operated continuously for 5 days when tested with water. This study demonstrates an important step toward ambulatory AL systems that automatically modulate ${\mathrm{CO}}_{2}$ removal as required by lung disease patients, thereby allowing for physiotherapy, comfort, and activity.

## 1. Introduction

Chronic obstructive pulmonary disease (COPD) ranks as a third leading cause of death worldwide and afflicts an estimated 328 million people [1]. As patients progress to end-stage lung disease (ESLD), their health and quality of life progressively worsen despite optimal care, medication, and supplemental oxygen. They become increasingly disabled, wheel-chaired then bed-ridden, and the dyspnea becomes so severe that patients cannot complete a sentence or swallow liquids [2]. Mechanical ventilation alleviates some of the symptoms, but the excessive oxygen concentrations and high airway pressures can induce pulmonary barotrauma, volutrauma, and biotrauma, exacerbating the original illness, and possibly resulting in multi-organ failure [3]. The only definitive treatment for ESLD is pulmonary transplantation, but the wait list is long, and many patients are not candidates due to age or comorbidities [4,5,6]. Artificial lung (AL) systems promptly relieve the symptoms of the ESLD without causing ventilator-induced injury [7,8,9,10]. These systems provide rest for the native lungs, allowing them to heal (bridge to recovery) or serve as a bridge to transplant by increasing odds of survival [11,12,13,14,15,16].

Ambulatory ALs have enabled rehabilitation through physical therapy, thereby reducing total hospital cost over treatment without physical rehabilitation [17]. Wearable artificial lung systems are currently under development [18,19,20]. However, no current clinical artificial lung system can respond to the changing respiratory needs of the patient. They thus risk removing too much carbon dioxide (${\mathrm{CO}}_{2}$) during periods of rest or improved lung function (causing hypocapnia and blood alkalosis) or not enough ${\mathrm{CO}}_{2}$ removal during physical activity or poor lung function (causing hypercapnia, dyspnea, and discomfort). Thus, activity, comfort, and especially rehabilitation are severely limited in current systems [21]. Various automatic controllers for AL systems have been investigated in the past [22,23,24,25]. Wartzek et al. investigated a fully automated ECMO system utilizing a ultrasonic flow sensor, automatically controlled blood pump, and an online blood gas analyzer [22]. Mendoza et al. studied an adaptive fuzzy logic controller based automated ECMO system in animal studies [23]. Conway et al. implemented a closed-loop feedback controller for an AL system using non-invasive sensors [24]. While successful, these experiments were implemented using components that would not permit their use in a long-term ambulatory application.

Hollow fiber membrane used in commercial ALs suffers from the problem of moisture accumulation on the gas phase, compromising their gas exchange ability when operational for prolonged periods [26]. Currently, such moisture is removed by manually increasing the sweep flow through the AL. The ability of the controller to automatically respond to changes in patient activity or disease status is critical for portable and wearable AL systems, as the hospital staff may not be available to adjust the respiratory support parameters in accordance with changes in patient activity. No current research or commercial AL system offers this capability.

Previous work by our laboratory has demonstrated that ${\mathrm{CO}}_{2}$ removal from blood in an AL is a nonlinear function of sweep air flow through the AL [27,28]. Based on this data, we previously built and evaluated a prototype benchtop, laptop-driven, wall-powered servoregulation controller that modulated sweep gas flow rate to respond automatically and rapidly to simulated metabolic changes [29]. Although successful, the design lacked components and packaging that would allow its use in long-term ambulatory applications. To overcome this drawback, this manuscript reports an upgraded design that implements a portable, battery-powered architecture with improvements in exhaust gas sampling and the addition of an automatic moisture removal feature from the AL and its testing and validation in vitro with whole blood and water. The controller has been designed with a focus on optimizing power consumption, reducing noise and vibration levels, and improving reliability to enable long-term operation.

## 2. Materials and Methods

### 2.1. Servoregulation Principle

The ${\mathrm{CO}}_{2}$-based sweep gas servoregulation principle used for the controller design is shown in Figure 1 [29]. The controller utilizes a proportional-integral-derivative (PID) principle-based closed-loop feedback mechanism to modulate blower-driven sweep air flow through an AL. The servoregulation controller operates in cycles, with each cycle being ∼2 s in length. In each cycle, current exhaust gas ${\mathrm{CO}}_{2}$ partial pressure EG${\mathrm{CO}}_{2}$ measured via a gas phase ${\mathrm{CO}}_{2}$ sensor is compared with the user-specified target EG${\mathrm{CO}}_{2}$ (tEG${\mathrm{CO}}_{2}$) value. The controller then adjusts sweep air flow through the AL depending on the error (EG${\mathrm{CO}}_{2}$-tEG${\mathrm{CO}}_{2}$) dynamics and tuning of proportional (P), integrative (I), and differential (D) parameters with the goal of eliminating the difference between measured and target EG${\mathrm{CO}}_{2}$.

### 2.2. Flow Path Design

A diagram of the gas flow path in the servoregulation controller is shown in Figure 2a. The components were selected with the goal of minimizing the portable system’s size, power consumption, and noise.

An air blower (WM7040-24V, Wei Cheng Electronics, Ningbo, China) intakes ambient air, providing up to 20 L/min of sweep flow through the AL at <45 dBA of noise and in a 70 × 70 × 40 mm form factor. Two blowers were integrated into the system for redundancy and fluidically combined via check valves to ensure unidirectional, leak-free airflow. The sweep flow rate is sensed via a flow sensor (SFM3400-AW, Sensirion AG, Stäfa, Switzerland) before being routed into the AL (Novalung iLA, Xenios AG, Heilbronn, Germany). Differential gas side pressure across the AL is measured via a pressure sensor (ABP2 series, Honeywell International Inc., Charlotte, NC, USA) and is used to monitor for a potential increase in flow resistance (due to moisture condensation). The exhaust gas leaving the AL is routed through a water trap before expiring into the atmosphere.

EG${\mathrm{CO}}_{2}$ was measured using side-stream capnography. A diaphragm pump (D250BLZ-V, TCS Micropumps, Kent, UK) continuously samples a small, fixed amount of the exhaust gas (∼200 mL/min) through a moisture trap (CM-0103, ${\mathrm{CO}}_{2}$-meter, Ormond Beach, FL, USA) and 24” of Nafion tubing (ME-110-24BB, Perma Pure LLC, Lakewood, NJ, USA) to remove condensation and reduce humidity which can otherwise affect ${\mathrm{CO}}_{2}$ sensor functionality. The dried air then enters the ${\mathrm{CO}}_{2}$ sensor assembly (Figure 2b) where it is heated by a 1-watt round Kapton polyimide heating strip before entering the sensing chamber integrating a ${\mathrm{CO}}_{2}$ sensor (SprintIR-W20%, ${\mathrm{CO}}_{2}$-meter, Ormond Beach, FL, USA). A temperature and humidity sensor (SHTC3, Sensirion AG, Stäfa, Switzerland) is used to monitor the sensing chamber temperature. A closed-loop PID feedback control is implemented using the heating strip in conjunction with the temperature and humidity sensor to maintain the sensing chamber temperature at 39 °C to increase the condensation point. Measured ${\mathrm{CO}}_{2}$ values are then compensated for barometric pressure by an absolute pressure sensor (ABP series, Honeywell International Inc., Charlotte, NC, USA). The aforementioned measures facilitate sampling of the exhaust gases at a relatively constant flow, pressure, and temperature, enabling accurate, reliable, uninterrupted measurement of EG${\mathrm{CO}}_{2}$ levels for long periods.

### 2.3. Supporting Electronics and Firmware

A Microcontroller development board (RM57L843, Texas Instruments, Dallas, TX, USA) interfaces with the various subsystems and sensors via a custom communication board and point-to-point wiring. The collected data is then logged into a laptop for analysis purposes.

A Graphical User Interface (GUI) is implemented using a 7” touchscreen (G2H2, Reach Technology, Lake Oswego, OR, USA). The touchscreen allows the user to configure settings and view current and recorded values, such as tEG${\mathrm{CO}}_{2}$, EG${\mathrm{CO}}_{2}$, flow rate, pressure, and PID settings.

The battery management system (Figure 3, blue blocks) consists of a 24 V 130 wh Li-ion battery pack (consisting of 7 series connected 21,700 cells), battery charger board, and AC to DC power source. The battery management system enables the servoregulator to operate on either battery or wall power. The battery power is regulated using a power conditioning circuitry (Figure 3, red blocks) consisting of an isolated 5 V voltage regulator that powers the microcontroller development board, sensing and control circuitry, and the touchscreen and an isolated 24 V voltage regulator that powers the blowers. The conditioning block isolates the system electronics and sensors from noise and faults on the battery side and protects the operator and patient from power surges when using wall power.

The system supports two modes of operation, fixed tEG${\mathrm{CO}}_{2}$ mode and fixed flow mode (similar to traditional AL controllers). In fixed tEG${\mathrm{CO}}_{2}$ mode, the PID controller modulates the blower’s speed to match the measured EG${\mathrm{CO}}_{2}$ with tEG${\mathrm{CO}}_{2}$. If the measured EG${\mathrm{CO}}_{2}$ is greater than tEG${\mathrm{CO}}_{2}$, the blower increases the sweep volume, increasing the ${\mathrm{CO}}_{2}$ removal rate, thus reducing EG${\mathrm{CO}}_{2}$. If the measured EG${\mathrm{CO}}_{2}$ is less than tEG${\mathrm{CO}}_{2}$, the blower reduces sweep flow, allowing EG${\mathrm{CO}}_{2}$ to rise. In fixed flow mode, the PID controller modulates the blower speed to achieve a target sweep gas flow rate. In each cycle, the current sweep flow is compared to the target sweep. If the recorded flow is higher than the user-defined flow, the controller reduces the power to the blower, and if the measured flow rate is less than the target flow rate, it increases the power to the blower. In both operating modes, the rate of convergence of measured value with target value is determined by the tuning of PID parameters. The PID controller is tuned using the GUI for both modes of operation.

### 2.4. In Vitro Testing with Blood and Water

The servoregulation controller was initially tested in vitro for ∼12 h using heparin-anticoagulated bovine blood from a local slaughterhouse (active clotting time ≥ 1000 s) to test the system’s ability to respond to changing blood partial pressure of ${\mathrm{CO}}_{2}$ (p${\mathrm{CO}}_{2}$) levels (Figure 4).

The test circuit is comprised of a main AL (Novalung iLA, Xenios AG, Heilbronn, Germany), with its gas ports connected to the servoregulation system. A Capiox RX25 oxygenator (Terumo Cardiovascular, Ann Arbor, MI, USA) with a maximum blood flow rate of 7.0 L/min was used as a blood conditioning lung with its gas ports connected to the gas sources (Figure 4). The blood flows in the test circuit were measured via a TS410 flow meter and ME-9PXL tubing flow probe (Transonic Systems Inc., Ithaca, NY, USA). Initially, while the servoregulator was off, blood in the circuit was allowed to reach the normal venous blood p${\mathrm{CO}}_{2}$ levels as defined by the US FDA guidelines using the conditioning lung at 37 °C [30]. Once blood conditioning was done, the servoregulator was turned on and the PID controller was grossly tuned using the Ziegler-Nichols method.

Initial tests were performed to evaluate the response time of the controller to the changes in blood p${\mathrm{CO}}_{2}$ levels. The blood flow through the main AL was adjusted to 0.5, 1.0 and 1.5 L/min while the tEG${\mathrm{CO}}_{2}$ was maintained at 20 mmHg. The PID controller was further tuned to achieve a quick response (<2 min) to changes in incoming blood p${\mathrm{CO}}_{2}$ levels with minimal oscillation. The inlet blood p${\mathrm{CO}}_{2}$ levels to the main AL that are a function of patient metabolic rate were changed to mimic the changes in patient activity by varying the ${\mathrm{CO}}_{2}$ concentration in the sweep gas flows through the conditioning AL. The blood in the test circuit was allowed to reach a steady state, as confirmed by the blood samples drawn at the inlet and outlet of the main AL and analyzed with a Radiometer ABL800 Flex (Brea, CA, USA) blood gas analyzer. The oxygen ($O_{2}$) delivery to the blood was calculated using blood hemoglobin (Hgb) levels that were measured at the interval of 60 min. The blood Hgb levels were measured as 13.7 ± 0.1 g/dL throughout the study.

The servoregulation controller was further tested for its ${\mathrm{CO}}_{2}$ removal and blood oxygenation ability. During these tests, the blood flow rates were changed between 0.5, 1.0 and 1.5 L/min while the tEG${\mathrm{CO}}_{2}$ was changed between 10, 20 and 40 mmHg. The blood p${\mathrm{CO}}_{2}$ levels at the inlet of the AL were modulated between 40 and 80 mmHg to challenge the controller at various clinically relevant blood p${\mathrm{CO}}_{2}$ levels [30]. Total 3 in vitro experiments were performed at distinct AL blood flows, with each experiment comprised of 9 servoregulation cycles (∼30 min each). Before every cycle, the servoregulation controller was kept off for approximately 15 minutes and sweep gas concentration through the conditioning AL was adjusted to modulate the p${\mathrm{CO}}_{2}$ level of the blood in the circuit allowing the blood to reach the target blood p${\mathrm{CO}}_{2}$ value. The controller was then turned on, thus allowing the sweep flow through the main AL (that corresponds to blood flow and tEG${\mathrm{CO}}_{2}$ setting) to remove ${\mathrm{CO}}_{2}$ from the blood in the circuit. The steady state blood-gas measurements that included pre and post AL blood p${\mathrm{CO}}_{2}$ levels (mmHg), pre and post AL blood oxygen saturation levels (%) were recorded after approximately 15 minutes. (i.e. once the blood p${\mathrm{CO}}_{2}$ level stabilized.) Servoregulation controller parameters such as steady state sweep flow rate (L/min), ambient pressure (mmHg) and power consumption (w) were recorded from the GUI once the steady state condition was reached.

The ${\mathrm{CO}}_{2}$ clearance (mL/min) by the servoregulation controller was calculated by,
(1)$$\begin{array}{c} \frac{\mathrm{steady}\;\mathrm{state}\;{\mathrm{EGCO}}_{2}\;\left(\mathrm{mmHg}\right)\times \mathrm{steady}\;\mathrm{state}\;\mathrm{sweep}\;\mathrm{flow}\;\left(L/\min\right)}{\mathrm{ambient}\;\mathrm{pressure}\;\left(\mathrm{mmHg}\right)\times 1000} \end{array}$$

The oxygen ($O_{2}$) delivery to the blood was calculated using blood hemoglobin (Hgb) levels that were measured at the interval of 60 min. The blood Hgb levels were measured as 13.7 ± 0.1 g/dL throughout the study. The oxygen delivery (cc/min) to the blood in the AL was calculated by [31],
(2)$$\begin{array}{c} \mathrm{blood}\;\mathrm{oxygen}\;\mathrm{saturation}\;(\%)\times \mathrm{blood}\;\mathrm{hemoglobin}\;\left(g/\mathrm{dL}\right)\\ \times \;\mathrm{steady}\;\mathrm{state}\;\mathrm{blood}\;\mathrm{flow}\;\left(L/\min\right)\times 0.136\;\left(\mathrm{cc}/g\right) \end{array}$$

The results are expressed as mean ± SD. Longitudinal data are plotted with respect to blood flow rate and tEG${\mathrm{CO}}_{2}$. The data analysis was performed in Microsoft Excel (2010) or MATLAB R2021b (The MathWorks, Inc., Natick, MA, USA) as required.

The servoregulator was also tested in vitro with distilled water for 5 days to confirm the robustness and reliability of the hardware, electronics, blowers, and firmware. The test circuit shown in Figure 4 was used for the 5-day tests, with water replacing blood in the flow circuit. An old AL that tends to accumulate moisture at the outlet was deliberately chosen for this study. Initial 5-day tests indicated that water accumulation in the gas phase caused an increase in resistance to sweep air flow and a degradation in ${\mathrm{CO}}_{2}$ removal ability of the AL [26]. Due to this effect, a feature was added in which the system monitors the gas-side resistance of the AL over time (using measured pressure and flow). When resistance was 1.55x baseline, the controller would automatically send a quick and large pulse (∼14 L/min) of sweep gas flow through the AL, clearing the condensation from the gas side until the resistance returned to near baseline (within 15%). With this new feature, subsequent 5-day runs were completed.

## 3. Results

### 3.1. In Vitro Testing with Blood

The portable servoregulation system integrates all components into a battery-driven 22 × 22 × 30 cm, 2 kg package (Figure 5). The battery pack enables servoregulation operation for ∼12 h on a full charge.

Data from an initial run at 1.0 L/min with bovine blood are shown in Figure 6. At time point ‘b’ (10:00 min), the inlet blood p${\mathrm{CO}}_{2}$ was increased to replicate an increase in patient metabolic rate by manually increasing the concentration of ${\mathrm{CO}}_{2}$ in the sweep gases to the conditioning AL (Figure 4). The resulting increase in the inlet blood p${\mathrm{CO}}_{2}$ in the main AL caused a brief increase in EG${\mathrm{CO}}_{2}$ (red dotted line, Figure 6). The PID controller rapidly responded to this change by increasing the blower-driven sweep flow to minimize the difference between tEG${\mathrm{CO}}_{2}$ and EG${\mathrm{CO}}_{2}$ (shown in the zoomed area). For this instance, blood p${\mathrm{CO}}_{2}$ was decreased from 89 mmHg (AL inlet) to 46 mmHg (AL outlet) while steady-state sweep flow was 6.5 L/min, for a tEG${\mathrm{CO}}_{2}$ of 20 mmHg.

A decrease in patient metabolic rate was simulated by decreasing the inlet blood p${\mathrm{CO}}_{2}$ as shown at time points ‘c’ (31:00 min) and ‘d’ (42:00 min) in Figure 6. The decrease in inlet blood p${\mathrm{CO}}_{2}$ caused a transient drop in EG${\mathrm{CO}}_{2}$. The controller quickly responded to this change by decreasing the sweep flow through the AL until EG${\mathrm{CO}}_{2}$ matched tEG${\mathrm{CO}}_{2}$ (20 mmHg) in both instances as shown in the zoomed images. For the change initiated at time point ‘c’, blood p${\mathrm{CO}}_{2}$ was decreased from 60 mmHg (inlet) to 40 mmHg (outlet) while steady-state sweep flow was 4.6 L/min. For the change initiated at time point ‘d’, blood p${\mathrm{CO}}_{2}$ was decreased from 47 mmHg (inlet) to 34 mmHg (outlet) while steady-state sweep flow was 3.4 L/min. For the remaining test duration, the controller was able to closely match the tEG${\mathrm{CO}}_{2}$ with the EG${\mathrm{CO}}_{2}$ value without oscillations.

Figure 7 shows combined data from 3 in vitro experiments with whole blood. (Raw data from the tests are shown in Appendix A). The performance of the servoregulation controller was initially evaluated for a constant tEG${\mathrm{CO}}_{2}$ of 20 mmHg while changing the device blood flow rates to 0.5, 1.0 and 1.5 L/min. Overall, the ${\mathrm{CO}}_{2}$ removal rate linearly increased with the blood flow rate for a constant tEG${\mathrm{CO}}_{2}$ as shown in Figure 7a. Then, the effects of tEG${\mathrm{CO}}_{2}$ levels on the ${\mathrm{CO}}_{2}$ removal were tested by changing the tEG${\mathrm{CO}}_{2}$ values to 10, 20 and 40 mmHg while maintaining the device blood flow rate at 1.0 L/min. The ${\mathrm{CO}}_{2}$ removal rate exhibited a mild inverse relationship with the tEG${\mathrm{CO}}_{2}$ level as shown in Figure 7b.

The performance of servoregulator was also evaluated for its ability to oxygenate the blood as shown in Figure 7c,d. Using air as the sweep gas, the system delivered between 22 and 66 mL/min of $O_{2}$ across all test conditions and increased blood ${\mathrm{sO}}_{2}$ between 11 and 45%. Overall, the $O_{2}$ delivery showed a correlation with the blood flow rate as shown in Figure 7c. Figure 7d shows that the blood oxygenation rate remained unchanged for the changes in tEG${\mathrm{CO}}_{2}$ (sweep flow) levels.

### 3.2. Long-Term In Vitro Testing with Water

Figure 8a shows the operation of the servoregulator from the 5-day in vitro test with water. The tEG${\mathrm{CO}}_{2}$ was set at 17 mmHg throughout the run while the water flow rate was maintained at 1.25 L/min. When the inlet water p${\mathrm{CO}}_{2}$ level was changed, the servoregulator compensated by adjusting sweep flow (blue) through the AL. The different levels of sweep flow values observed are the outcome of the changes in inlet water p${\mathrm{CO}}_{2}$ levels. The inlet water p${\mathrm{CO}}_{2}$ levels were changed between 35 and 70 mmHg by changing the ${\mathrm{CO}}_{2}$ concentration in the sweep gases through the conditioning AL. The servoregulator responded to every such change by changing the sweep flows through the main AL to maintain the outlet water p${\mathrm{CO}}_{2}$ levels between 15 and 17 mmHg.

ALs accumulate moisture at the outlet of the fiber bundles as moisture from the blood side diffuses through the polymer membrane, becoming water vapor on the intraluminal gas phase [26]. That moisture then condenses in the gas phase of the AL due to the temperature gradient between the blood and gas phases of the AL. As the system continues to operate, the condensate obstructs more fibers, reducing the effective membrane surface area for gaseous exchange, thus reducing the ${\mathrm{CO}}_{2}$ clearance ability of the ALs over time. Automated and rapid sweep gas flow pulses are used to blow the moisture out of the fiber bundle and to re-establish regular gas exchange. The intermittent spikes in the sweep gas flow (blue) observed in Figure 8a are the result of the gas flushing process.

Figure 8b shows the profiles of sweep gas flow rate, power consumption, and AL gas-side pressure drop from 51:00 to 61:00 h. The servoregulator continues to monitor the increase in pressure drop (green) for a sweep flow (blue) until reaching its trigger value (point ‘a’). The gas flushing process initiated removal of the moisture from the AL circuit, which is evident from the reduction in pressure drop across the AL (green) for the similar sweep flow value (blue). When servoregulation is re-initiated, the power consumption (black) is significantly reduced compared to its pre-flushing value (∼25% drop). As the system operates, moisture condensation continues as shown by the slow increase in the pressure drop for a fixed sweep flow rate. The servoregulator increases the power input to the blowers to nullify the effects of this additional resistance until the gas flushing removes the moisture, as shown in Figure 8b at time points ‘a’, ‘b’, ‘c’ and ‘d’.

## 4. Discussion

The development of an automated controller that can support changing metabolic needs of the patients on respiratory support is critical for the development of ambulatory AL systems. The presented portable servoregulation controller advances previous work from our laboratory [29] in terms of miniaturized/integrated components to reduce size and weight; addition of centrifugal blowers instead of diaphragm pumps to enable operation at lower flows, and reduced noise; addition of a battery pack and charging system; introduction of a gas sampling system to fix sampling rate and achieve consistency of PID operation; addition of nafion tubing and heated ${\mathrm{CO}}_{2}$ chamber instead of a desiccant based moisture removal technique to eliminate consumables and implementation of a flushing system that automatically removes condensation in the AL to enable long-term operation.

During in vitro testing with whole blood, the controller was able to respond to the changes in the blood p${\mathrm{CO}}_{2}$ levels (<2 min) by rapidly modulating the sweep flows through the AL. The selected values of target EG${\mathrm{CO}}_{2}$ should allow for adequate ${\mathrm{CO}}_{2}$ removal to maintain blood ${\mathrm{CO}}_{2}$ levels to normal value during the conditions of normocapnia and hypercapnia. Overall, ${\mathrm{CO}}_{2}$ removal showed an inverse relationship with the target EG${\mathrm{CO}}_{2}$ and a direct correlation with the blood flow rate. For all variations in the EG${\mathrm{CO}}_{2}$ values and blood flow rates, the controller was able to maintain adequate sweep flow to oxygenate the blood in the AL.

Given the limitation of hollow fiber lung technology to accumulate moisture at the end of the fiber bundle, the addition of a sweep gas flushing mechanism to reduce moisture condensation is critical to sustaining the ${\mathrm{CO}}_{2}$ removal ability of the AL in the long run. Due to this reason, the in vitro test with water was deliberately performed using an old AL, which was prone to accumulate moisture at a higher rate, to challenge the sweep gas flushing system. The results show that the sweep gas flushing system was able to successfully monitor and flush away condensation in the fiber bundles of the AL and thereby support the ${\mathrm{CO}}_{2}$ removal operation of the AL for the entire test duration. This is a key component in developing a fully automated ambulatory AL system for long-term use.

Side-stream capnography is used for the measurement of the exhaust gas ${\mathrm{CO}}_{2}$ levels. While proven effective during the in vitro studies with blood and water, the ability of the utilized setup to respond to the changes in EG${\mathrm{CO}}_{2}$ was observed to be hindered at the lower sweep flow rates (<0.5 L/min) due to the limitation of the sampling pump to overcome the resistance caused by the length of sampling circuit and moisture accumulation in the AL. This problem was overcome by keeping a minimum blower-driven sweep flow above 0.5 L/min.

The success of a future ambulatory AL system used in the clinical setting will be dependent on the effectiveness of the moisture removal and gas sampling system. Future work involves further miniaturization of the presented work into a wearable system where efforts will be dedicated to reducing sampling time delays to improve the overall response time of the controller.

## 5. Conclusions

A portable sweep gas servoregulation controller for the AL systems that uses closed-loop feedback control of ${\mathrm{CO}}_{2}$ in the exhaust gases to respond to the variation in blood p${\mathrm{CO}}_{2}$ levels has been described. The presented system design advances previous work from our lab by miniaturizing the system’s size and weight, and adding an exhaust gas sampling system as well as an automated moisture removal system. The in vitro studies with blood successfully demonstrated the ${\mathrm{CO}}_{2}$ management capability of the controller while maintaining satisfactory blood oxygenation levels. The in vitro studies with water demonstrated that the controller could maintain the gas exchange performance of the AL over a longer duration by successfully removing the moisture accumulated at the outlet of the AL. This work provides a basis for the development of long-term ambulatory AL systems that can respond to the changing metabolic needs of ESLD patients, thus allowing for increased physical activity and better quality of life for such patients.

## Figures and Tables

**Figure 1 bioengineering-09-00593-f001:**
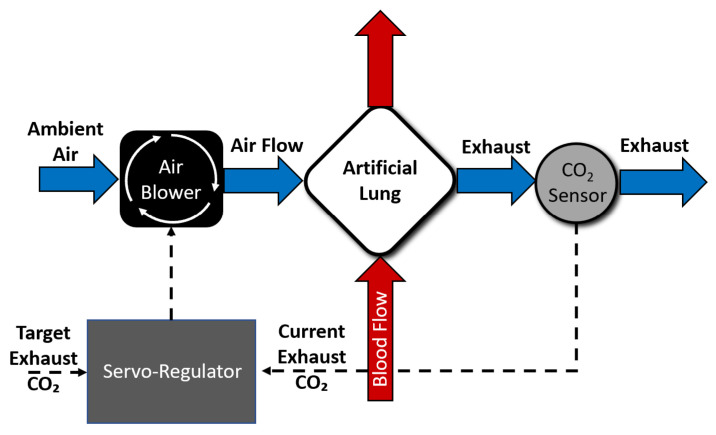
An overview of servoregulation-based sweep gas control.

**Figure 2 bioengineering-09-00593-f002:**
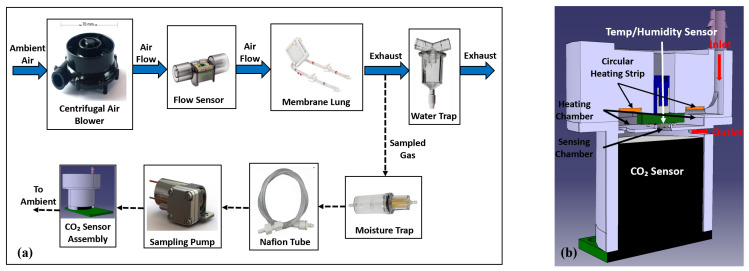
(**a**) Flow Diagram of servoregulator with images of major system components (**b**) Cut section view of the ${\mathrm{CO}}_{2}$ sensor assembly.

**Figure 3 bioengineering-09-00593-f003:**
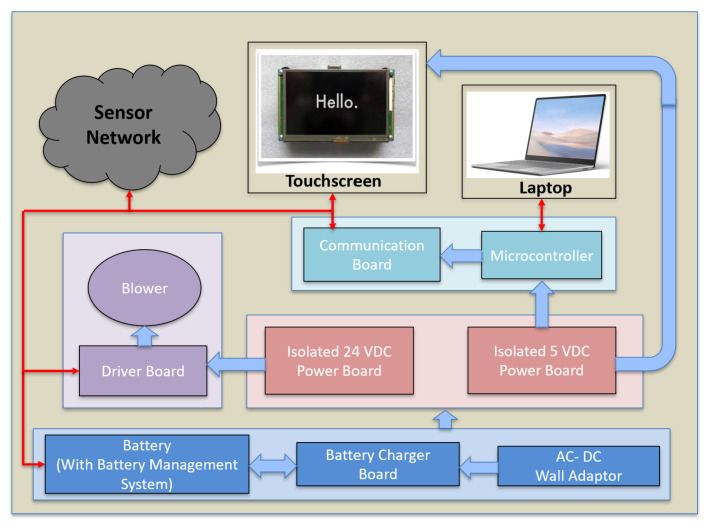
The architecture of the servoregulator with main power and electronics components labeled. The solid blue arrows represent power lines while the red arrows depict communication lines.

**Figure 4 bioengineering-09-00593-f004:**
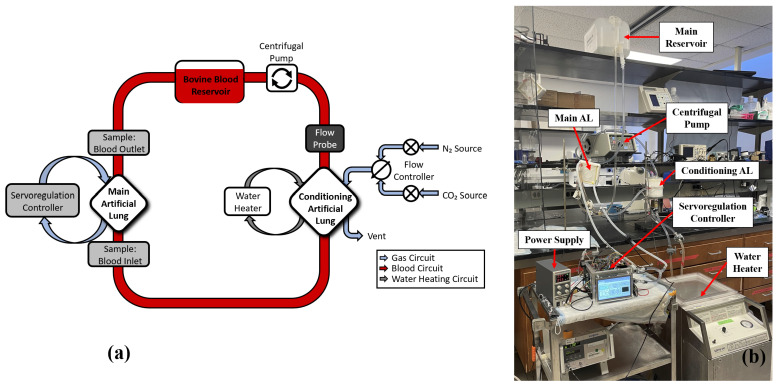
(**a**) Figure representing in vitro test circuit. Bovine blood in the circuit was replaced with distilled water during long-term testing. (**b**) An image showing the in vitro testing setup with main components labeled.

**Figure 5 bioengineering-09-00593-f005:**
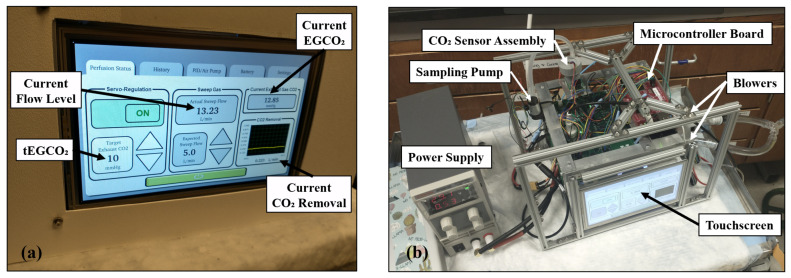
Photographs showing portable servoregulation controller (**a**) with enclosure and (**b**) without the enclosure.

**Figure 6 bioengineering-09-00593-f006:**
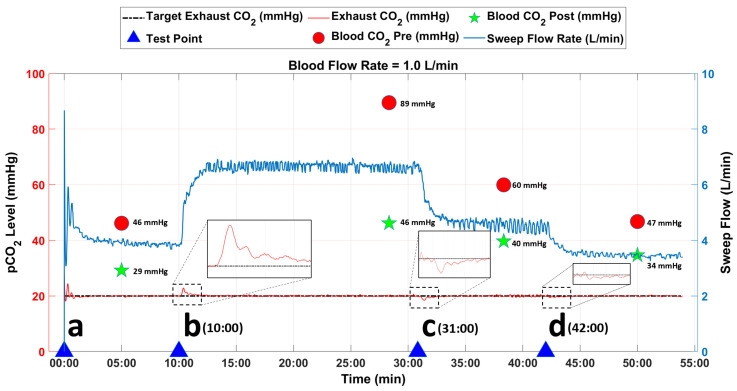
Data from in vitro study at AL blood flow rate of 1.0 L/min. The blue arrows represent the time points when the inlet blood p${\mathrm{CO}}_{2}$ levels were changed to challenge the tuning of the PID controller.

**Figure 7 bioengineering-09-00593-f007:**
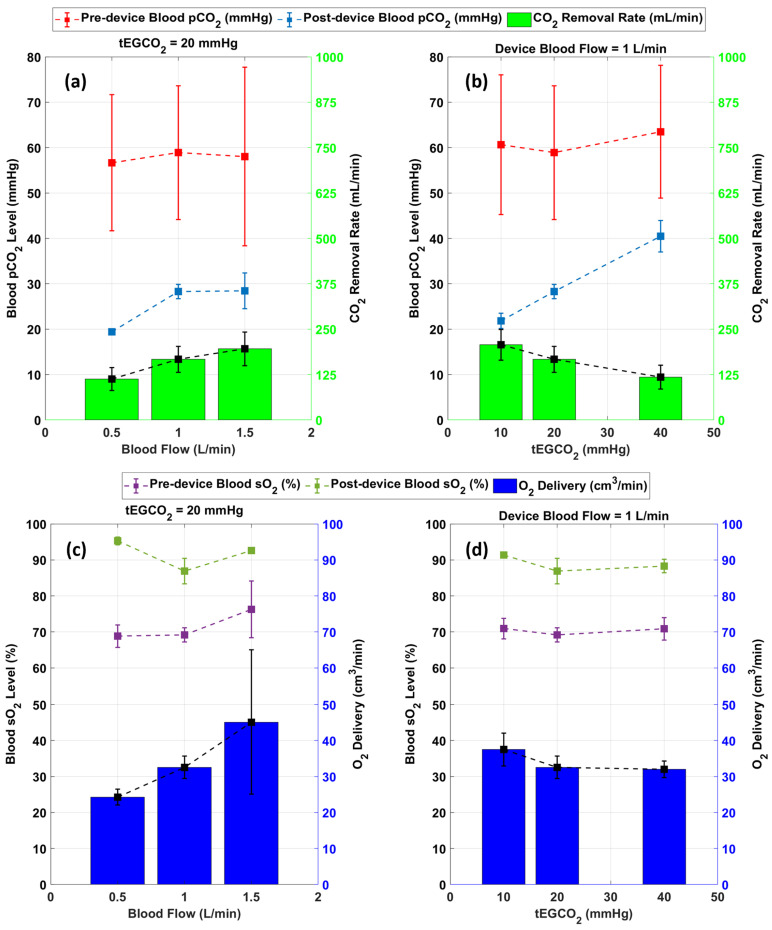
Result summary of in vitro studies showing (**a**) Effect of blood flow on ${\mathrm{CO}}_{2}$ removal rate (**b**) effect of tEG${\mathrm{CO}}_{2}$ on ${\mathrm{CO}}_{2}$ removal (**c**) $O_{2}$ delivery at various blood flows (**d**) $O_{2}$ delivery at various tEG${\mathrm{CO}}_{2}$ levels (*n* = 3).

**Figure 8 bioengineering-09-00593-f008:**
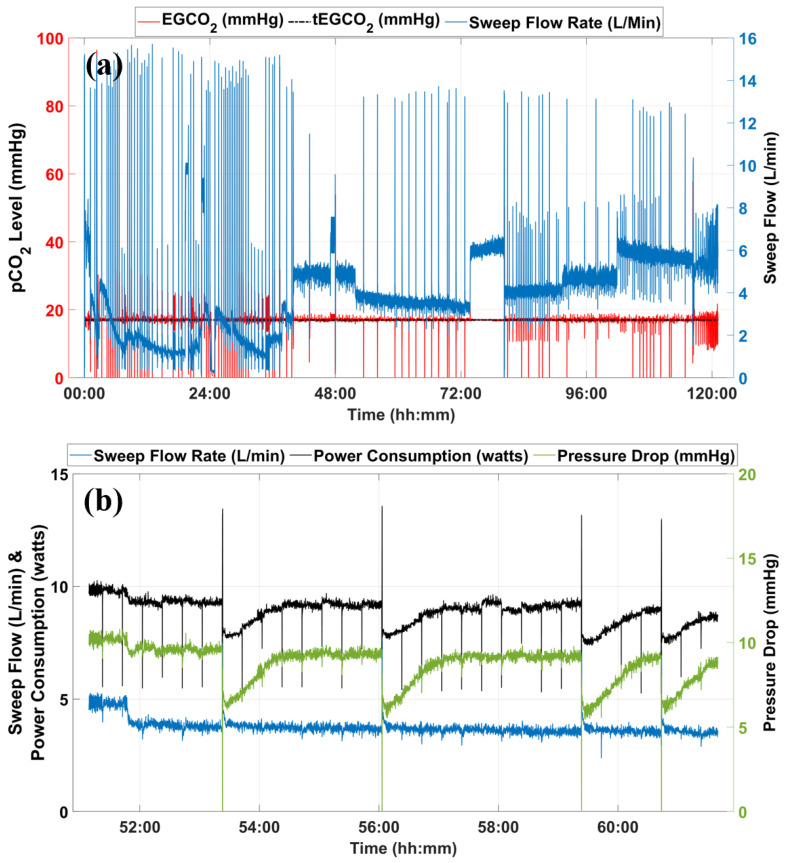
(**a**) Data from long-term in vitro testing using an old AL with water at the flow rate of 1.25 L/min. (**b**) Snippet showing sweep flow rate, system power consumption, and pressure drop across the AL for system operation between 51:00 to 61:00 h.

## Data Availability

The data presented in this study are available in Appendix A.

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
