# Peer review of "A Portable Servoregulation Controller to Automate CO2 Removal in Artificial Lungs"

_bioengineering, 2022, doi:10.3390/bioengineering9100593_

Round 1
Reviewer 1 Report
The article entitled A Portable Servoregulation Controller to Automate CO2
Removal in Artificial Lungs is a document of interesting subject matter.
1. Your abstract should clearly state the essence of the problem you are addressing, what you did and what you found and recommend. That will help a prospective reader of the abstract to decide if they wish to read the entire article
2. It is expected to have an extensive literature review followed by an in-depth and critical analysis of the state of the art, and identify challenges for future research in the Introduction.
3. The authors should do the analysis the conclusion section must clearly establish a strong correlation with the proposed topic.
4. The objective or objectives should be clearly elucidated in the last paragraph of the introduction.
Reviewer 2 Report
The authors presented the design and implementation of an improved servoregulation controller for the automatic adjustment of CO2 levels in artificial lungs.
Quality of English needs to be improved as there are grammar mistakes and typos throughout the manuscript.
Materials and Methods
- How long was each cycle of the servoregulation controller?
- The grammar of the sentence “pressure EGCO2 is measured via a CO2 sensor is …” does not seem to be correct
- What was the minimum size of the components of the controller?
- Labels in Figure 2 are too small and in Fig. 2b some of the white labels are not readable because they are on a very clear background
- In section 2.3, “(Fig. ??)” should be substituted with the actual number of the figures
- The two pictures in Fig. 4 should be labelled (a) and (b) instead of referring to them as “left” and “right”
- The symbol “&” is used a few times in the text. It should be changed into “and”
- Figure 5 is very unclear because it is very difficult to see clearly and read the screen shown in figure (a). In addition, Figure (b) is quite small and is difficult to clearly see the different components. The black arrows used in figure (b) are not clearly visible because they are placed on a very dark background
- How many experiments were conducted?
- What type of analysis was conducted on the data collected?
Results
- Figure 6: It is not clear what time the blue arrows indicate. It would be better to label those times with the actual values. The red points and green stars do not show with sufficient clarity what values they refer to. The lines for “target exhaust CO2” and “exhaust CO2” are not visible with sufficient clarity
- Use “and” instead of “&”
- In the text, the exact times corresponding to “point ‘b’”, “point ‘c’”, etc should be given
- What does “pCO2” stand for? Explain when used for the first time
- Specify what “clinically-relevant blood pCO2 levels” is the text referring to
- In section 3.1, “Fig. ??” should be substituted with the actual number of the figure
- Figure 8 is very confusing. The graphs are too small. In figure (a) the red dataset is almost entirely covered by the clue one; and in figure (b) it is unclear whether there are three or two labels on the x-axis. It would be advisable to organise the graphs vertically rather than horizontally
Discussion
The section is very short and does not seem to refer to previous versions of the same apparatus for the purpose of discussing and further supporting the claim of improved performance of the controller. This would make the discussion stronger.
